# Clinical Characteristics, Management, and Prognostic Factors of Appendiceal Neuroendocrine Neoplasms: Insights from a Multicenter International Study

**DOI:** 10.3390/biomedicines13112724

**Published:** 2025-11-06

**Authors:** Federica Cavalcoli, Kasun Samarasinghe, Alessandro Del Gobbo, Niall Mulligan, Emanuele Rausa, Alberto Caimo, Paolo Cantù, Gianluca Tamagno, Sara Massironi

**Affiliations:** 1Gastroenterology and Digestive Endoscopy Unit, Fondazione IRCCS Istituto Nazionale dei Tumori, Via Venezian 1, 20133 Milan, Italy; paolo.cantu@istitutotumori.mi.it; 2Department of Endocrinology-Diabetes Mellitus, Mater Misericordiae University Hospital, Dublin-Co. Dublin 7, Dublin, Ireland; gianlucatamagno@tiscali.it; 3Department of Medicine, Wexford General Hospital—University College Dublin, Wexford-Co, Y35 Y17D Wexford, Ireland; 4Pathology Department, Fondazione IRCCS Ca’ Granda Ospedale Maggiore Policlinico, 20135 Milan, Italy; alessandro.delgobbo@policlinico.mi.it; 5Department of Pathology, Mater Misericordiae University Hospital, Dublin-Co. Dublin, Dublin 7, Dublin, Ireland; 6Unit of Hereditary Digestive Tract Tumors, Fondazione IRCCS Istituto Nazionale dei Tumori, 20133 Milan, Italy; 7School of Mathematical Sciences, Technological University Dublin, Dublin-Co. Dublin, TU910 Dublin, Ireland; 8Blackrock Health, Blackrock Clinic and Hermitage Clinic, Dublin-Co. Dublin, Dublin 20, Dublin, Ireland; 9Department of Medicine and Surgery; Vita-Salute San Raffele University, 20132 Milan, Italy; sara.massironi@libero.it

**Keywords:** appendiceal neuroendocrine neoplasm, appendiceal neuroendocrine tumor, appendiceal neuroendocrine carcinoma, carcinoid, appendix, appendectomy

## Abstract

**Introduction:** Appendiceal neuroendocrine neoplasms (aNENs) are the most common malignant appendiceal neoplasms. Localized aNENs are typically managed with an appendectomy; however, right colectomy may be necessary in patients with a high risk of nodal disease. However, the role of right hemicolectomy and the optimal surveillance strategy, particularly for tumors between 1 and 2 cm, remains controversial. **Material and Methods:** This retrospective, observational study evaluated patients diagnosed with aNENs between January 1995 and July 2015 at two tertiary centers in Ireland and Italy. Data were extracted from a prospectively maintained registry and included clinical, pathological, and therapeutic variables, as well as follow-up outcomes. **Results:** Forty-three patients (41.8% male; median age 27.5 years) were included, with a median follow-up of 49 months. The median tumor size was 6.4 mm (range: 0.6–40 mm). The majority were G1 tumors (58%), and staging distribution was predominantly Stage I (60%). While no significant differences in demographics or tumor features were observed between centers, completion right hemicolectomies were more frequent in the Irish cohort (*p* = 0.04). Follow-up practices varied, with more intensive imaging and biochemical monitoring observed in the Italian cohort. Overall prognosis was excellent, with a single case of recurrence during the study period. **Conclusions:** Most aNENs are effectively managed with appendectomy alone, and routine follow-up may be unnecessary in the absence of adverse pathological features. Accurate risk stratification, driven by comprehensive histopathological assessment, is critical for optimizing management and surveillance strategies.

## 1. Introduction

Appendiceal neuroendocrine neoplasms (aNENs) represent the most frequent subtype of appendiceal malignancies, accounting for approximately 50–77% of all appendiceal tumors [1]. These lesions are typically discovered incidentally during appendectomy procedures performed for suspected acute appendicitis, with an estimated incidence of 0.3% to 0.9% among appendectomy specimens [1,2,3,4,5]. Although considered rare, recent epidemiological studies suggest that the incidence of aNENs is increasing, potentially reflecting improved detection methods, enhanced pathological classification systems, and broader use of advanced imaging technologies [5,6,7,8,9].

Most aNENs are diagnosed in young adults between 20 and 50 years of age, with a slight female predominance [2,5], although occurrence in children and adolescents has also been reported [10].

Due to the absence of specific clinical symptoms and the frequent incidental nature of their diagnosis, preoperative identification remains challenging [11]. 

From a pathological standpoint, aNENs are usually small, well-differentiated (G1), low-proliferation tumors (Ki-67 <3%), most often confined to the tip of the appendix and discovered incidentally. They show minimal atypia and rarely present with lymphovascular invasion or nodal metastases. In contrast, colonic NENs (especially those from the right colon) tend to be larger, higher-grade tumors and are more frequently associated with nodal and distant metastases. Rectal NENs, while often well-differentiated, carry a higher metastatic risk once they exceed 10 mm. Ileal NETs, on the other hand, are often multifocal, strongly associated with mesenteric fibrosis, lymph node involvement, and serotonin secretion. They typically present at a more advanced stage, despite being well-differentiated, and require long-term follow-up even after radical surgery. These anatomical and biological differences support distinct management strategies: while appendectomy is often sufficient for low-risk aNENs, more aggressive surgical and surveillance approaches are necessary for colonic, rectal, and ileal NETs.

The prognosis of aNENs is generally excellent, with a 5-year survival rate close to 100% for localized disease and between 85% and 100% for locoregional disease [1,6,12,13]. Tumor size, location, lymphovascular invasion, and the extent of mesoappendiceal invasion have been identified as risk factors for lymph node involvement and distant metastases. However, cases of aggressive disease with distant metastases have been sporadically reported even for aNENs smaller than 1 cm [12]. 

Management of aNENs traditionally depends on tumor size, grade, and the presence of histopathological risk factors such as mesoappendiceal invasion, lymphovascular invasion, positive margins, or a high proliferative index [2,14]. While simple appendectomy is generally sufficient for tumors smaller than 1 cm without risk features, controversy persists regarding the optimal treatment for tumors measuring 1–2 cm. Recent large-scale studies have questioned the necessity of right hemicolectomy in this subgroup, demonstrating no significant survival advantage compared to appendectomy alone, particularly in low-grade tumors [2,6,14,15,16,17,18]. Accordingly, Nesti et al., in a multicentric European study, suggested that right-sided hemicolectomy is not indicated after complete resection of 1–2 cm aNENs by appendectomy, and that regional lymph node metastases do not impact survival [2,19].

The role of systematic pathological assessment is pivotal in guiding management. Incomplete pathology reporting—especially regarding mesoappendiceal invasion, lymphovascular involvement, and Ki-67 proliferation index—can result in inappropriate therapeutic decisions [2,20]. Consequently, international guidelines now advocate for synoptic pathology reporting to ensure the comprehensive evaluation of key prognostic variables [1]. 

Despite excellent overall survival rates, nodal metastases and rare instances of recurrence have been reported, particularly in tumors with adverse histopathological features [5,15]. However, consensus on the indications for extended surgery and standardized surveillance protocols remains lacking [2,14,16,21,22,23]. In this context, the present study aims to retrospectively analyze the clinical characteristics, pathological features, management strategies, and outcomes of patients with aNENs at two European tertiary centers. Our secondary objective is to highlight areas of clinical uncertainty and heterogeneity in the diagnostic and therapeutic approach to aNENs, with the goal of informing and optimizing real-world management strategies.

## 2. Materials and Methods

### 2.1. Study Design and Setting

This study is a retrospective, observational analysis conducted at two tertiary referral centers: Fondazione IRCCS Ca’ Granda Ospedale Maggiore Policlinico (Milan, Italy) and Mater Misericordiae University Hospital (Dublin, Ireland). Institutional registries of neuroendocrine neoplasms were queried to identify patients diagnosed with appendiceal neuroendocrine neoplasms (aNENs) between January 1995 and July 2015.

The study period was limited to cases diagnosed before July 2015 to ensure adequate follow-up and consistency in the availability of critical pathological and clinical variables. The study was conducted in accordance with the principles of the Declaration of Helsinki.

Inclusion criteria were a histologically confirmed well-differentiated aNEN, age over five years, absence of synchronous malignancies, and a minimum follow-up duration of six months. Exclusion criteria included tumors of non-neuroendocrine origin, poorly differentiated neuroendocrine carcinomas, goblet cell tumors (due to their distinct biological behavior), and cases with incomplete clinical or pathological data.

### 2.2. Data Collection

Clinical, demographic, and pathological data were extracted from prospectively maintained databases and supplemented through retrospective chart review when necessary. Collected variables included: age, sex, clinical presentation (including whether the diagnosis was incidental), type of surgical intervention (appendectomy alone or right hemicolectomy), histopathological features (tumor size, grade, margin status, mesoappendiceal invasion, lymphovascular invasion, perineural invasion), postoperative surveillance protocols, and clinical outcomes including recurrence, metastasis, and survival status. Tumor grading and staging were performed according to the World Health Organization (WHO) classification of digestive system tumors [24] and the American Joint Committee on Cancer (AJCC) 8th edition staging criteria [25].

### 2.3. Outcomes

The primary outcomes were overall survival (OS) and recurrence-free survival (RFS). Secondary outcomes included the incidence of nodal or distant metastasis and the necessity for surgical re-intervention.

### 2.4. Statistical Analysis

Statistical analyses were conducted using SPSS version 27.0 (IBM Corp., Armonk, New York, USA). Continuous variables were expressed as medians with interquartile ranges (IQRs) or means with standard deviations (SDs), according to data distribution. The Shapiro–Wilk test was used to assess normality. Categorical variables were reported as absolute numbers and percentages. Group comparisons were performed using the Chi-square or Fisher’s exact test for categorical variables, and the Mann–Whitney U test or Student’s t-test for continuous variables, as appropriate. A two-sided *p*-value < 0.05 was considered statistically significant.

## 3. Results

During the study period (January 1995 to July 2015), a total of 4011 appendectomies were performed across the two participating centers. Among these, 58 patients were diagnosed with aNENs based on histopathological examination. After applying the exclusion criteria, 43 patients were included in the final analysis (Figure 1).

Key clinical and pathological variables, including tumor size, grading, staging, management strategies, and outcomes, are summarized in Table 1 and Table 2. 

Twenty-two patients were from the Irish center (10 males and 11 females, median age: 26 years) and 21 were from the Italian center (8 males and 13 females, median age: 27.5 years).

Specimens for histological examination were available for all patients. The median tumor size was 6.4 mm (range: 0.6–40 mm). Histological examination revealed a well-differentiated neuroendocrine tumor in all cases, classified as grade 1 (*n* = 25, 58% of cases), or grade 2 (four patients, 30%). Ki-67 index was not documented in 14 cases (32.6%). Mitotic count data were reported in 18 to 21 patients, with no mitotic figures observed in 11 cases, 1 mitotic figure per 10 high-power fields (HPF) in 3 cases, and 2 mitotic figures per 10 HPF in another 3 cases. Vascular invasion data were retrieved for 23 out of 43 patients (53.5%), showing absence of vascular invasion in 22 cases. These gaps reflect historical variability in pathology reporting practices. 

According to TNM staging at the time of diagnosis, 26 patients (60.5%) were classified as having stage I disease, 14 patients (32.6%) had stage II disease, and 3 patients (6.9%) had stage III disease. Among stage III patients, two cases showed serosal invasion, while one case demonstrated lymph node metastasis. No significant differences were observed between the Italian and Irish cohorts with respect to patient age at diagnosis, tumor size, gender distribution, Ki-67 index, or tumor staging.

Regarding surgical management, initial surgery consisted of appendectomy in 39 of 43 patients (90.7%). The main indications for surgery were acute appendicitis in 38 cases (88.4%), followed by bowel obstruction in three cases (7%) and inflammatory bowel disease in two cases (4.6%). One patient who presented with obstructive bowel symptoms underwent ileocecal resection and lymphadenectomy, which revealed nodal involvement on pathological examination. Notably, none of the patients presented with symptoms suggestive of carcinoid syndrome, such as flushing, diarrhea, or other peptide-mediated symptoms.

Overall, 11 patients (25.6%) underwent re-intervention with right hemicolectomy. The decision for completion surgery was predominantly based on adverse pathological features, including tumor size greater than 2 cm in two cases, mesoappendiceal invasion in eight cases and positive surgical margins in one patient.

Right hemicolectomy revealed lymph node metastasis in two patients. Surgical re-intervention with right hemicolectomy following the initial appendectomy was significantly more common among patients managed at Mater Misericordiae University Hospital compared to Fondazione IRCCS Ca’ Granda (*p* = 0.04) (Figure 2). The follow-up protocols varied significantly between the two institutions. Patients managed in Italy underwent more intensive surveillance, with a median follow-up duration of 24 months (range: 6–140 months), including periodic clinical assessments and imaging as per institutional protocols. In contrast, Irish patients had a median follow-up of 8 months (range: 6–60 months), with a less structured approach primarily based on symptomatic evaluation. However, the median duration of follow-up difference was not statistically significant. Detailed data on surveillance modalities are reported in Table 2. While the overall number of patients undergoing follow-up assessments did not differ significantly between the two cohorts (Figure 3), the Italian cohort underwent abdominal ultrasound, somatostatin receptor imaging, and biochemical surveillance significantly more frequently than the Irish cohort (Figure 4 and Figure 5). In contrast, the use of CT and MRI was comparable between the two groups (Figure 4). 

After a median follow-up of 49 months (interquartile range: 21–78 months), the overall prognosis was excellent. No disease-related deaths were recorded during the study period. 

In the present series 10 patients had aNENs between 1 and 2 cm, 8 in the Irish court and two in the Italian. In this subset, the median age of diagnosis was 26.5 years (7 females and 3 males). Histological examination showed a well-differentiated neuroendocrine tumor in all cases, with grade 1 in five cases (50% of cases), grade 2 in two patients (20%), while ki-67 index was not documented in 3 cases (30%). Regarding to TNM staging, 4 patients (40%) had stage I disease at the time of diagnosis, 5 patients (50%) had stage II disease, and 1 patient (10%) had stage III disease.

Overall, 4 patients (58.3%) underwent re-intervention with right hemicolectomy mostly due to adverse pathological features.

Only one patient developed disease recurrence. This was a 51-year-old female who was initially diagnosed with a 20 mm well-differentiated G1 aNEN (Ki-67 1%) confined to the appendiceal wall. She presented with obstructive symptoms, and the initial treatment consisted of an ileocecal resection with lymphadenectomy. Following the histopathological diagnosis of NEN, completion of right hemicolectomy was not recommended after multidisciplinary team discussion. The patient was subsequently monitored with annual abdominal CT scans, somatostatin receptor scintigraphy, and clinical evaluations. However, 24 months after surgery, disease progression was detected, with hepatic, mesenteric, and locoregional metastases. After further multidisciplinary evaluation, the patient commenced treatment with somatostatin analogues and remained alive at the last follow-up, 27 months after the initial surgery.

Due to the limited number of events reported, DFS and OS were not reached during the observation period. Comparative analysis revealed no statistically significant differences in recurrence-free survival or overall survival between patients managed by appendectomy alone versus those undergoing right hemicolectomy. Furthermore, tumor size, grade, and the presence of mesoappendiceal invasion did not demonstrate a significant association with recurrence within this cohort.

## 4. Discussion

This retrospective, multicenter study analyzed the clinical presentation, pathological features, management strategies, and outcomes of patients with aNENs across two European tertiary centers.

Consistent with previous literature [2,5,8,11,26,27], most tumors were diagnosed incidentally during surgery for suspected acute appendicitis, reflecting the non-specific clinical presentation and limited diagnostic sensitivity of preoperative imaging. The demographic and pathological profile of our cohort aligns with existing data, showing a predominance of young adult patients, small tumor size, low Ki-67 indices [26,28], well-differentiated grade 1 neoplasms [5], and a slight, non-significant of female predominance [7,28,29]. 

Most tumors were located at the tip of the appendix and classified as Stage I at diagnosis, confirming the typically indolent behavior of aNENs [14,15].

Surgical management of aNENs remains debated, particularly regarding the need for right hemicolectomy in tumors between 1 and 2 cm [1,14].

In our series, most patients underwent appendectomy alone, with secondary surgery performed based on adverse pathological features. We observed no significant difference in recurrence-free or overall survival between patients undergoing appendectomy versus right hemicolectomy, supporting a conservative surgical approach [14,30,31]. Specifically, given the ongoing controversy surrounding the optimal management of 1–2 cm aNENs, we performed a focused sub-analysis of this subgroup (*n* = 10, representing approximately 23% of the cohort). Consistent with prior large-scale studies [2,14,18,19,31], no recurrence or disease-specific mortality was observed in this size range, regardless of whether patients underwent appendectomy alone or completion right hemicolectomy. Although the small number of cases precludes statistical comparison, these findings support the growing consensus that appendectomy may be sufficient for well-differentiated 1–2 cm aNENs without additional high-risk features. However, larger multicenter datasets are required to confirm these observations. 

The 2023 ENETS guidelines recognize that 1–2 cm tumors without high-grade features may be managed with appendectomy alone [2,3,24], suggesting that completion right hemicolectomy may be unnecessary in an increasing number of cases [19]. Although nodal or distant metastases can occur in tumors with risk factors [6], our data suggest that regional lymph node involvement does not significantly affect prognosis, reinforcing the notion that regional lymphatic involvement may not impact clinical outcomes [30]. This aligns with pediatric and young adult studies reporting near 100% survival despite nodal disease [30].

Pathology plays a crucial role in risk stratification, particularly for incidental aNENs [32,33,34,35,36]. Incomplete pathological evaluations can lead to mismanagement, especially in 1–2 cm tumors. Recent ENETS guidelines emphasize synoptic reporting to ensure documentation of tumor size, localization, grading, extension, vascular and lymphatic invasion, and TNM staging [2]. 

We observed no significant differences in age or treatment between Irish and Italian patients. However, surveillance protocols varied significantly, with more intensive monitoring in the Italian cohort. Despite these differences, outcomes were favorable, suggesting intensive follow-up may be unnecessary for low-risk cases [22,37]. Our findings support risk-adapted surveillance strategies to minimize unnecessary procedures and reduce healthcare costs [5,38].

Of note, one Irish patient developed locoregional and hepatic metastases 24 months post-surgery. While ENETS guidelines suggest nodal involvement does not increase recurrence risk, this case underscores the need for tailored follow-up in selected patients. Current ENETS Guidelines [1], based on different international studies [14,15,19,30,36,39,40,41,42,43,44,45] suggest that lymph node involvement in aNENs is not associated with an increased risk of recurrence, distant metastasis, or disease-specific mortality, thereby supporting less aggressive surgical and surveillance strategies. However, most studies reporting these outcomes have relatively limited follow-up durations. The case reported herein highlights a rare instance of aNEN exhibiting aggressive biological behavior, with the development of metachronous liver metastases two years after diagnosis. This experience suggests that nodal disease may indeed confer a higher risk of recurrence and supports the need for a carefully tailored follow-up strategy in patients with high risk factors.

Current international guidelines recommend that no follow-up is necessary for aNENs smaller than 1 cm in diameter with radical (R0) resection. Conversely, long-term follow-up is advised for patients with lymph node involvement, tumors between 1 and 2 cm with additional risk factors who did not undergo right hemicolectomy, or patients with initial distant metastases [2]. Nevertheless, the optimal timing, duration, and modalities of follow-up remain undefined, and no universally accepted protocol currently exists.

The findings from the present series reinforce the concept that most aNENs exhibit indolent behavior, consistent with international data [15,30,46,47]. Thus, in the absence of pathological risk factors and with exhaustive pathology reporting, routine follow-up may not be necessary for low-risk patients. However, in cases with lymph node involvement deeper mesoappendiceal invasion, or other high-risk features, both more aggressive surgical management and structured follow-up are justified, given the potential risk of disease recurrence. 

Considering the financial costs and psychological burden associated with prolonged follow-up, particularly for younger patients, optimizing patient selection for surveillance is essential. In this setting, a multidisciplinary approach, involving surgeons, oncologists, pathologists, and radiologists, is recommended to ensure individualized decision-making and to balance the risks and benefits of intensive monitoring [5,38,48,49].

This study has different limitations, including its retrospective design, relatively small sample size, and variations in histopathological reporting between institutions. The small cohort and the occurrence of only a single recurrence event render the analysis underpowered to draw definitive conclusions on surgical outcomes. Therefore, while our findings align with the current literature supporting conservative management, they should be interpreted with caution. Moreover, the substantial proportion of missing pathological data, particularly regarding mesoappendiceal invasion, vascular invasion, and Ki-67 index, further limits the robustness of the risk stratification analysis and underscores the need for standardized synoptic pathology reporting, as recommended by current ENETS guidelines [2]. Our findings further emphasize the importance of standardizing histopathological assessment to enhance data comparability across retrospective cohorts.

Furthermore, although our follow-up duration is comparable to other published series, longer-term data would provide further reassurance regarding late recurrences, which, although rare, have been occasionally described [13,38]. Moreover, the heterogeneity in follow-up protocols between centers represents a potential source of surveillance bias. Again, this variability underscores the necessity for standardized follow-up algorithms, as outlined in recent ENETS recommendations. Future multicenter studies should prioritize harmonization of surveillance strategies to minimize such inconsistencies.

Nonetheless, our results add to the growing body of evidence suggesting that a tailored approach to aNENs is warranted. Our data suggest that appendectomy alone may be sufficient for selected patients with tumors measuring 1–2 cm, particularly in the absence of adverse histopathological features. However, we acknowledge that the small sample size and variability in follow-up protocols limit the strength of this conclusion. Therefore, we advocate for individualized management decisions, guided by multidisciplinary evaluation. The development of internationally harmonized guidelines, particularly for tumors between 1 and 2 cm, remains a critical unmet need [30].

## 5. Conclusions

This multicenter retrospective study confirms that aNENs are predominantly indolent tumors with excellent long-term prognosis when appropriately managed. Appendectomy alone appears sufficient for most cases, particularly for small, well-differentiated lesions without adverse pathological features. Our findings, in line with current international guidelines, support a selective approach to right hemicolectomy, reserving more aggressive surgery for patients with significant histopathological risk factors.

While surveillance strategies remain heterogeneous, our data suggest that routine intensive follow-up may not be necessary for low-risk patients with complete resection. However, the occurrence of late recurrence in a patient with nodal involvement highlights that structured surveillance should still be considered for selected high-risk cases. Given the potential psychological and financial burden of prolonged follow-up, patient selection must be optimized through multidisciplinary team discussions. Future efforts should focus on establishing standardized, risk-adapted guidelines for surgical management and follow-up of aNENs, aiming to further personalize care while minimizing overtreatment.

## Figures and Tables

**Figure 1 biomedicines-13-02724-f001:**
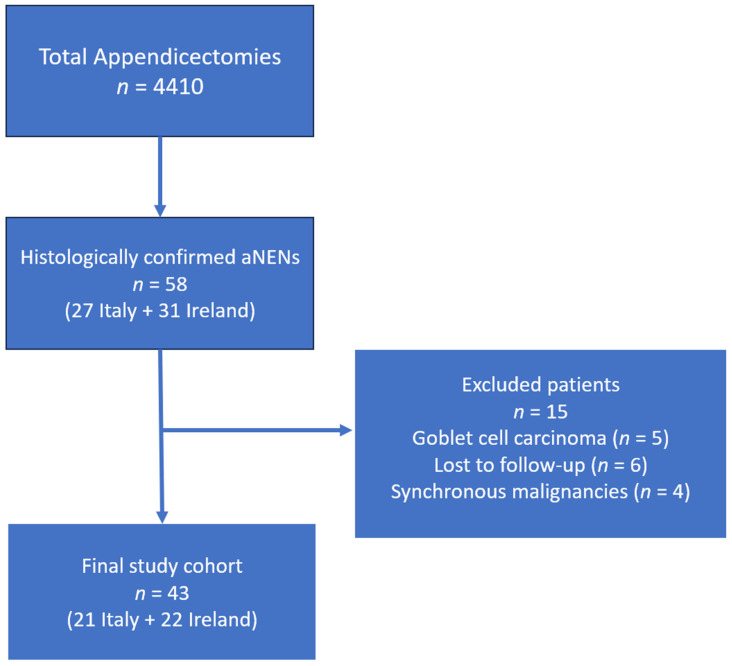
PRISMA Flowchart depicting the study cohort selection process. The diagram summarizes the number of appendectomies performed, patients diagnosed with aNENs, exclusions (e.g., incomplete data, synchronous malignancies), and the final cohort analyzed (*n* = 43).

**Figure 2 biomedicines-13-02724-f002:**
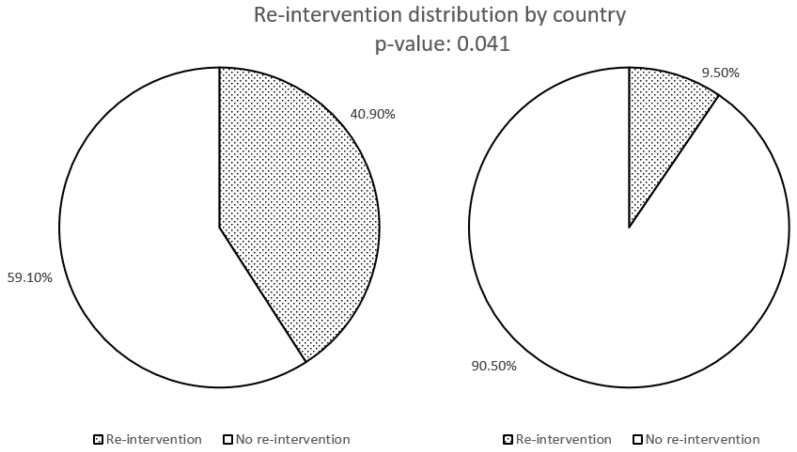
Number of patients who underwent re-intervention after initial surgery.

**Figure 3 biomedicines-13-02724-f003:**
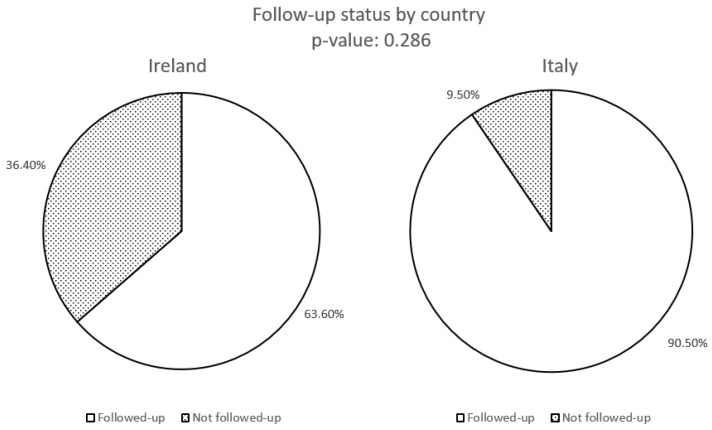
Number of patients who underwent follow-up after initial surgery.

**Figure 4 biomedicines-13-02724-f004:**
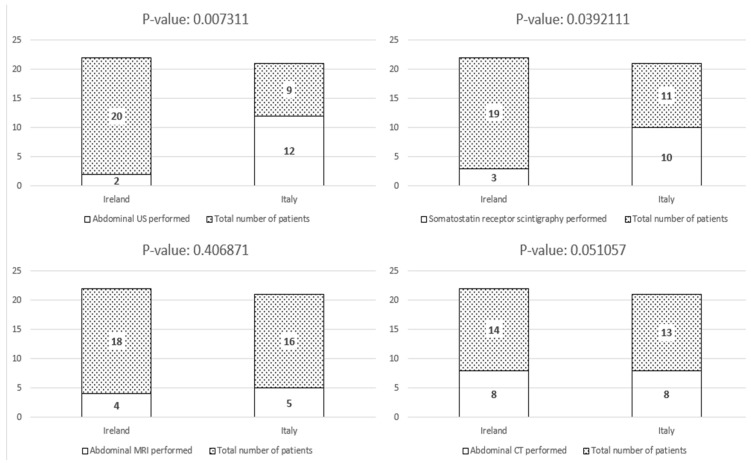
Number of patients who underwent imaging (abdominal US, somatostatin receptor scintigraphy, abdominal CT or abdominal MRI) during follow-up.

**Figure 5 biomedicines-13-02724-f005:**
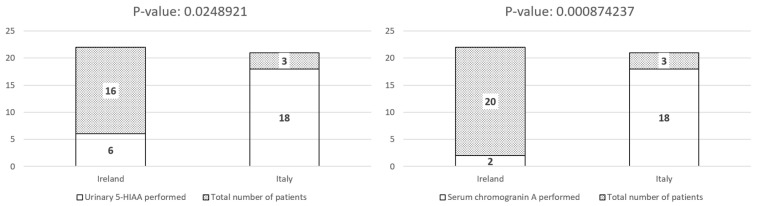
Number of patients who underwent biochemical testing for urinary 5-HIAA or chromogranin A during follow-up.

**Table 1 biomedicines-13-02724-t001:** Demographic features, tumor classification, and re-intervention rate of appendiceal Neuroendocrine Neoplasm (aNEN) patients in the Irish and in the Italian cohort.

	Ireland (*n* = 22)	Italy (*n* = 21)
Median age (range)	28 years (14–63 years)	31 years (9–75 years)
Female/Male	12/10	13/8
Median tumor size (range)	10 mm (3–25 mm)	5 mm (0.6–40 mm)
aNEN stage [23] I II III	12 9 1	14 5 2
aNEN grade G1 G2 G3	13 3 0	12 1 0
Tumor location	Apex: 9 (40.9%) Base: 2 (9.1%) Not specified: 11 (50.0%)	Apex: 12 (57.1%) Base: 1 (4.8%) Not specified: 8 (38.1%)
Mesoappendiceal invasion	Present: 7 (31.8%) Absent: 8 (36.4%) Not reported: 7 (31.8%)	Present: 4 (19.0%) Absent: 11 (52.4%) Not reported: 6 (28.6%)
Initial surgery	Appendectomy: 20 (90.9%) Ileocecal resection: 1 (4.5%) Right hemicolectomy: 1 (4.5%)	Appendectomy: 19 (90.5%) Total colectomy: 1 (4.8%) Right hemicolectomy: 1 (4.8%)
Re-intervention with right hemicolectomy	9 *	2 *

* Statistically significant: *p*-value < 0.05. *n*: Number of patients; G1: Grade 1 (Ki-67 index <3%); G2: Grade 2 (Ki-67 index 3–20%); G3: Grade 3 (Ki-67 index >20%); Stage I/II/III: Tumor staging according to AJCC TNM classification [23]; mm: Millimeters.

**Table 2 biomedicines-13-02724-t002:** Follow-up strategy for appendiceal neuroendocrine neoplasm (aNEN) patients in the Irish and in the Italian cohort.

	Ireland (*n* = 22)	Italy (*n* = 21)
Followed-up patients N (%)	14 (63.6%)	19 (90.5%)
Duration of the follow-up months (range)	8 (6–60 months)	24 (6–140 months)
Outpatient clinics visit N (%)	13 (59.1%)	18 (85.7%)
Abdominal US N (%)	2 (9.1%) *	12 (57.1%) *
Abdominal CT N (%)	8 (36.4%)	8 (38.1%)
Abdominal MRI N (%)	4 (18.2%)	5 (23.8%)
Somatostatin receptor scintigraphy N (%)	3 (13.6%) *	10 (47.6%) *
^68^Gallium PET/CT N (%)	0 *	2 (9.5%) *
Urinary 5-HIAA N (%)	6 (27.3%) *	18 (85.7%) *
Serum chromogranin A N (%)	2 (9.1%) *	18 (85.7%) *

* Statistically significant: *p*-value < 0.05. *n*: Number of patients; US: Ultrasound; CT: Computed Tomography; MRI: Magnetic Resonance Imaging; 68Gallium PET/CT: Positron Emission Tomography/Computed Tomography using 68Ga-DOTATATE tracer; 5-HIAA: 5-Hydroxyindoleacetic Acid.

## Data Availability

The data presented in this study are available on request from the corresponding author.

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
