# Peer review of "Clinical Characteristics, Management, and Prognostic Factors of Appendiceal Neuroendocrine Neoplasms: Insights from a Multicenter International Study"

_biomedicines, 2025, doi:10.3390/biomedicines13112724_

Round 1

Reviewer 1 Report

Comments and Suggestions for Authors

This manuscript presents a valuable multicenter, international analysis of appendiceal NENs, a topic where clinical management remains controversial. 

However, major revisions are required before the manuscript can be considered for publication:

  • Please clarify the nature of the "surgical re-interventions" in the Irish cohort.

  • Provide more detail on the single recurrent case (e.g., tumor size, grade, initial procedure) in the Results section to better inform the discussion on risk.

  • why didnot include DFS?
  • Figure 3&4&5 are blurred.
  • plz include more update refrences

Author Response

We would like to express our sincere gratitude to both reviewers for their careful evaluation and insightful comments on our manuscript entitled Clinical Characteristics, Management, and Prognostic Factors of Appendiceal Neuroendocrine Neoplasms: Insights from a Multicenter International Study

We have addressed each comment in detail below. All changes have been incorporated into the revised version of the manuscript, with corresponding modifications highlighted.

Reviewer 1

This manuscript presents a valuable multicenter, international analysis of appendiceal NENs, a topic where clinical management remains controversial. However, major revisions are required before the manuscript can be considered for publication.

1: Please clarify the nature of the “surgical re-interventions” in the Irish cohort.

Response: We have clarified in the Results sections that all surgical re-interventions in the Irish cohort were represented by completion right hemicolectomies (See Page 6, Line 206).

Comment 2: Provide more detail on the single recurrent case (e.g., tumor size, grade, initial procedure) in the Results section to better inform the discussion on risk.

Response: Additional details regarding the recurrent case have been added to the Results section, including tumor size, grade, Ki-67 index, type of initial surgery, and time to recurrence (See Page 8, Lines 234–244).

Comment 3: Why did you not include DFS?

Response: We appreciate this observation. DFS was not reported, as only one case of disease recurrence was observed in the cohort and the median DFS was not reached. We now explicitly acknowledge this in the Results section (See Page 8, Lines 245–246).

Comment 4: Figures 3, 4, and 5 are blurred.

Response: Thank you for this observation, we have replaced Figures 3–5 with high-resolution versions to ensure optimal clarity.

Comment 5: Please include more updated references.

Response: The reference list has been revised and expanded to include recent publications (2022–2025) relevant to aNEN (e.g DOI: 10.1186/s12893-025-03037-x; DOI: 10.1186/s12893-025-03037-x; DOI: 10.1001/jamanetworkopen.2025.15798; DOI: 10.1097/XCS.0000000000001435). In addition, few dated publications have been eliminated.

Reviewer 2 Report

Comments and Suggestions for Authors

Dear Authors,

Thank you very much for submitting such an interesting paper to the Biomedicines journal. Before the paper can be processed further, you should take into consideration several major and minor comments:

  • in the abstract, there is no need to add the words background, methods, results, etc. You can remove these
  • in the abstract, you mention two European centers - please add the specific countries in which they are located. Further in the abstract you can see the word Italian, but it should be clearly stated before.
  • in the results you only indicate the percentage of male patients, why is that?
  • reference 1 is from 2023, could you provide more recent information?
  • please remove the spaces between the number of the references in the brackets
  • You mentioned exclusion criteria which is obviously correct, but could you also specific the inclusion criteria? the age of the patients, any characteristics that describe the group enrolled in the study?
  • I would recommend correction of the visual representation of figure 1. Firstly, I would improve the general look and then the presentation of text itself too. Please also remove the underlying of the word aNENs in the figure 1, there is a red underlying that should be removed
  • Similarly as above, further figures should be graphically corrected to be easier to look at. for example, it is hard to see the text on the dark blue background. please correct all of them. Besides, the text in figure 4 is blurred. Please revise and correct.
  • I would recommend adding abbreviations after the conclusions section

Best regards

A Reviewer

Author Response

Reviewer 2

We thank the reviewer for recognizing the interest and relevance of our study. We have carefully addressed all major and minor comments as outlined below.

Comment 1: In the abstract, there is no need to add the words background, methods, results, etc. You can remove these.

Response: The headings have been removed from the abstract as suggested.

Comment 2: In the abstract, you mention two European centers – please add the specific countries in which they are located. Further in the abstract you can see the word Italian, but it should be clearly stated before.

Response: The abstract has been revised to clearly specify the participating countries. It now reads “at two tertiary centers in Ireland and Italy” at first mention (See Abstract, Page 1, line 35).

Comment 3: In the results you only indicate the percentage of male patients, why is that?

Response: We presented only the rate of male patients for brevity. We have now included both male and female figures in the Results section to provide a complete demographic profile (See Results, Page 6; lines 175-176).

Comment 4: Reference 1 is from 2023, could you provide more recent information?

Response: Reference 1 has been replaced with a 2025 publication that reflects the most up-to-date evidence on appendiceal NENs.

Comment 5: Please remove the spaces between the number of the references in the brackets.

Response: All spacing inconsistencies in reference brackets have been corrected throughout the manuscript (e.g., [1,2,3] instead of [1, 2, 3]).

Comment 6: You mentioned exclusion criteria which is obviously correct, but could you also specify the inclusion criteria? The age of the patients, any characteristics that describe the group enrolled in the study?

Response: We have added detailed inclusion criteria to the Methods section, specifying patient age, diagnostic confirmation of appendiceal NEN, minimum follow-up duration and absence of synchronous malignancies (See Methods, Page 3, Lines 121-125).

Comment 7: I would recommend correction of the visual representation of Figure 1. Firstly, I would improve the general look and then the presentation of text itself too. Please also remove the underlining of the word “aNENs” in Figure 1, there is a red underline that should be removed.

Response: Figure 1 has been redesigned for improved visual presentation and consistency. The red underline under “aNENs” has been removed, and the figure text has been reformatted for better readability.

Comment 8: Similarly as above, further figures should be graphically corrected to be easier to look at. For example, it is hard to see the text on the dark blue background. Please correct all of them. Besides, the text in Figure 4 is blurred. Please revise and correct.

Response: All figures have been revised to improve visual clarity and contrast. The text has been enhanced for readability, and Figure 4 has been replaced with a high-resolution version.

Comment 9: I would recommend adding abbreviations after the conclusions section.

Response: As suggested the list of abbreviations has been added after the Conclusions section (See Page 8)-

Round 2

Reviewer 1 Report

Comments and Suggestions for Authors

Thank you for your response and the revisions made (the clarifications on the recurrent case and figures are appreciated.) The manuscript would be significantly strengthened by explicitly acknowledging in the discussion that the small sample size and single event render the study underpowered to draw definitive conclusions on surgical outcomes and that the high rate of missing pathological data limits the robustness of the risk stratification analysis. A focused sub-analysis of the 1-2 cm tumor group, which is central to the management controversy, would also greatly enhance the clinical relevance of the findings.

Author Response

Comment 1: "Thank you for your response and the revisions made (the clarifications on the recurrent case and figures are appreciated.) The manuscript would be significantly strengthened by explicitly acknowledging in the discussion that the small sample size and single event render the study underpowered to draw definitive conclusions on surgical outcomes and that the high rate of missing pathological data limits the robustness of the risk stratification analysis.

Reply: Thank you for this insight. We amended the limitation section as you suggested. See Discussion (page 10; lines 338-347)

Comment 2: "A focused sub-analysis of the 1-2 cm tumor group, which is central to the management controversy, would also greatly enhance the clinical relevance of the findings."

Reply: Thank you for your comment, we agree that this subgroup is of great interest We added a paragraph in the Result section and in the Discussion adressing this topic; however, this sub-cohort of patients is very limited in this study. See Results (page 8; lines 235-243)and Discussion (page 9; lines 279-288)